# BRAIN-INSPIRED CONTINUAL PRE-TRAINED LEARNER VIA SILENT SYNAPTIC CONSOLIDATION

## ABSTRACT

Pre-trained models have demonstrated impressive generalization capabilities, yet they remain vulnerable to catastrophic forgetting when incrementally trained on new tasks. Existing architecture-based strategies encounter two primary challenges: Firstly, integrating a pre-trained network with a trainable sub-network complicates the delicate balance between learning plasticity and memory stability across evolving tasks during learning. Secondly, the absence of robust interconnections between pre-trained networks and various sub-networks limits the effective retrieval of pertinent information during inference. In this study, we introduce the *Artsy framework*, inspired by the activation mechanisms of silent synapses via spike-timing-dependent plasticity observed in mature biological neural networks, to enhance the continual learning capabilities of pre-trained models. The Artsy framework integrates two key components: 1) During training, the framework mimics mature brain dynamics by maintaining memory stability for previously learned knowledge within the pre-trained network while simultaneously promoting learning plasticity in task-specific sub-networks. 2) During inference, artificial silent and functional synapses are utilized to establish precise connections between the pre-synaptic neurons in the pre-trained network and the post-synaptic neurons in the sub-networks, facilitated through synaptic consolidation, thereby enabling effective extraction of relevant information from test samples. Comprehensive experimental evaluations reveal that our model significantly outperforms conventional methods on class-incremental learning tasks, while also providing enhanced biological interpretability for architecture-based approaches. Moreover, we propose that the Artsy framework offers a promising avenue for simulating biological synaptic mechanisms, potentially advancing our understanding of neural plasticity in both artificial and biological systems.

## 1 INTRODUCTION

Pre-trained artificial neural networks have demonstrated notable generalization capabilities; however, they are prone to catastrophic forgetting when exposed to sequential training on new datasets, as outlined in previous studies Wang et al. (2024). To overcome this limitation, it is imperative for pre-trained models to employ continual learning (CL) strategies that enable them to assimilate new tasks while preserving previously acquired knowledge. The process of training artificial networks using various CL methodologies is depicted in Figure 1. Within the sequential learning paradigm, both regularization-based Kirkpatrick et al. (2017) and optimization-based approaches Lopez-Paz & Ranzato (2017); Zeng et al. (2019) have been developed to mitigate memory degradation, attempting to maintain previous knowledge through weight regularization and gradient projection. Nonetheless, these methods frequently encounter difficulties in striking an optimal balance between learning plasticity for new tasks and maintaining stability of the established knowledge. In contrast, within the joint learning framework, replay-based strategies Shin et al. (2017); Van de Ven et al. (2020) sidestep the plasticity-stability trade-off by replaying experiences, utilizing generative models, and leveraging features from previously learned data. However, this approach often incurs substantial energy costs, particularly when training large-scale models on extensive raw datasets, as seen in models like GPT-3. Alternatively, in the split learning paradigm, architecture-based methods Aljundi et al. (2017); Yu et al. (2024) employ modular networks to facilitate learning from new data, offering a more energy-efficient solution.

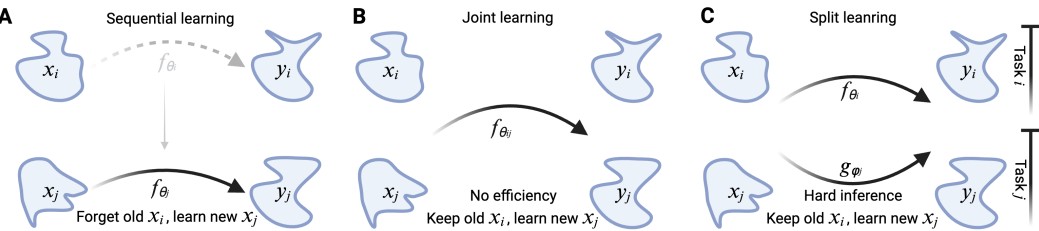

Figure 1: Illustration of artificial networks ($f$ and $g$) trained using various learning methods on a continual learning task $j$ where $i < j$. Initially, network $f$ is trained via back-propagation on task $i$. The goal of continual learning is to sequentially learn the subsequent task $j$. (A) With $f$ fixed, the parameters $\theta_i$ are optimized to $\theta_j$ on task $j$, enabling generalization to task $j$. (B) Keeping $f$ fixed, the parameters $\theta_i$ are optimized to $\theta_{ij}$ on tasks $i$ and $j$, allowing generalization to both tasks $i$ and $j$. (C) With $f$ fixed at parameters $\theta_i$, an additional sub-network $g$ with parameters $\varphi_j$ is introduced to generalize to both tasks $i$ and $j$.

Nevertheless, architecture-based approaches encounter two fundamental challenges. Firstly, during the training phase, integrating a frozen pre-trained network with a trainable sub-network (e.g., Adaptformer Chen et al. (2022) or Mixture-of-Experts Jacobs et al. (1991)) necessitates optimizing lightweight parameters within the sub-network Zhou et al. (2024a); Yu et al. (2024) to accommodate a range of new tasks. This integration presents a considerable challenge in maintaining an optimal balance between learning plasticity for novel tasks and memory stability for previously acquired knowledge, akin to the issues faced in sequential learning paradigms. Secondly, in the testing phase, the model, comprising a static pre-trained network and several trained sub-networks that encapsulate specialized knowledge for various tasks Zhou et al. (2024b), lacks an effective mechanism for establishing connections between these networks. Consequently, the absence of learned inter-network pathways hinders the model's ability to effectively extract and leverage relevant information from test samples, thereby limiting prediction accuracy.

Biological neural networks exhibit remarkable proficiency in addressing CL tasks, a capability fundamentally rooted in synaptic consolidation mechanisms. Synaptic plasticity plays a crucial role in achieving a balance between learning plasticity and memory stability, particularly in the developing, prepubescent brain. As young organisms (both animals and humans) acclimate to novel environments, synaptic plasticity drives dynamic alterations in the synaptic connections between presynaptic and postsynaptic neurons, facilitating adaptation to environmental changes Kerchner & Nicoll (2008); Hanse et al. (2013). Silent synapses, in particular, contribute to this adaptive capacity by enabling highly flexible connectivity between neurons Durand et al. (1996); Huang et al. (2015), thereby supporting synaptic plasticity. Studies by Liao et al. (1995) and Isaac et al. (1995) have identified the presence of silent synapses in the hippocampus, characterized by the mediation of N-methyl-D-aspartate (NMDA) receptor responses in the absence of $\alpha$-amino-3-hydroxy-5-methyl-4-isoxazole propionic acid (AMPA) receptor-mediated activity. The structural configuration of these silent synapses is depicted in Figure 2. Moreover, the process of synaptic conversion—whereby silent synapses become functional (unsilencing) and vice versa—occurs frequently during brain development but is considered less common in mature neural networks Hanse et al. (2013). Despite this, the adult brain retains the capacity for learning plasticity and memory stability, largely owing to the ability of silent synapses located at filopodia to transform into functional synapses, thereby forming new synaptic connections Vardalaki et al. (2022). Additionally, recent evidence Vardalaki et al. (2022) suggests that the formation of new synaptic connections in the adult neocortex is instrumental not only in preserving existing memories via mature synapses at dendritic spines but also in facilitating the rapid acquisition of new information through the dynamic activity of silent synapses at filopodia.

In this study, we propose the Artsy framework, a novel approach to enhancing the CL capabilities of pre-trained models by leveraging insights from the activation mechanisms of silent synapses via spike-timing-dependent plasticity observed in biological neural networks. This biological process, which enables the conversion of silent synapses into functional synapses within the adult neocortex Vardalaki et al. (2022), serves as a foundation for expanding the learning potential of mature neural systems. The Artsy framework is designed around two key components: (1) It emulates the mature

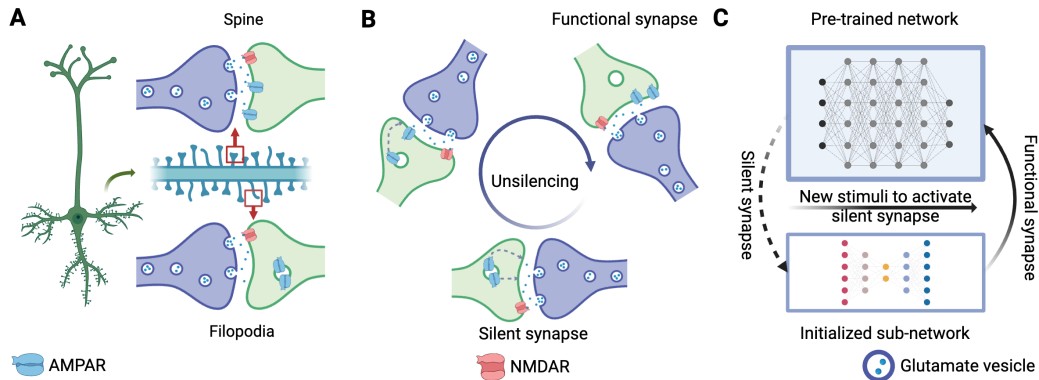

Figure 2: Overview of the connections between a pre-trained network and the initialized sub-networks via artificial silent and functional synapses. (A) In the mature brain, dendritic segments comprise silent synapses located at filopodia and functional synapses at dendritic spines. An archetypal glutamatergic synapse consists of presynaptic and postsynaptic membranes. The presynaptic terminal contains glutamate-filled vesicles, while the postsynaptic membrane contains AMPA and NMDA receptors. (B) The process of converting silent synapses into functional synapses through AMPA receptor unsilencing. AMPA unsilencing involves the synaptic incorporation of clusters of AMPA receptors, converting silent synapses into functional ones, whereas AMPA silencing entails the loss of synaptic AMPA receptor clusters. (C) In the Artsy framework, artificial silent and functional synapses connect the pre-trained network to the initialized sub-network. New stimulus inputs can induce the conversion of artificial silent synapses into functional synapses.

brain by integrating memory stability for previously acquired knowledge within the pre-trained network with learning plasticity for new data in the initialized sub-networks during the training phase; (2) Artificial silent and functional synapses form dynamic connections between pre-synaptic neurons in the pre-trained network and post-synaptic neurons in the sub-networks via synaptic consolidation, thereby enabling the extraction of relevant information from test samples during the inference phase.

In this framework, functional synapses require both AMPA and NMDA receptors to facilitate connections, whereas silent synapses possess only NMDA receptors, rendering them functionally inactive. Upon exposure to novel stimuli processed by pre-synaptic neurons within the pre-trained network, pre-synaptic activity—mimicking glutamate uncaging and spike-timing-dependent plasticity—triggers the insertion of AMPA receptors into previously silent synapses. Simultaneously, current injection into the soma of the post-synaptic neuron within the initialized sub-network generates an action potential, effectively converting silent synapses into functional ones. These artificial synaptic connections are initialized through a trainable network, as detailed in Section 3. By adopting this synaptic consolidation mechanism, the Artsy framework demonstrates the ability to continuously learn new tasks while preserving previously acquired memories, closely mirroring the adaptive capabilities of the mature brain.

To the best of our knowledge, this study is the first to introduce the Artsy framework, inspired by the role of silent synapses at filopodia in the mature brain, to enhance the neural plasticity of pre-trained networks through the integration of an initialized sub-network via artificial synapse consolidation. We have implemented the Artsy framework on class-incremental learning tasks, and extensive experimental evaluations demonstrate that our model not only outperforms conventional methods but also provides a more biologically interpretable approach compared to existing architecture-based methods, which often lack clear biological explanations.

This paper is organized as follows: We begin by providing a concise overview of class-incremental learning and the concept of silent synapses (Section 2). Subsequently, we detail how these silent synapse mechanisms are translated into the Artsy framework for class-incremental learning tasks (Section 3). Following this, we present our experimental results, showcasing the continual learning capabilities of the Artsy framework in comparison with relevant state-of-the-art benchmarks (Section 4). Finally, we engage in a discussion of related work (Section 5) and provide a conclusion (Section 6).

## 2 BACKGROUND ON CONTINUAL LEARNING AND SILENT SYNAPSE

In this section, we present an overview of the foundational concepts in continual learning and silent synapses. While continual learning encompasses various task types Van de Ven et al. (2022), our study specifically targets class-incremental learning (CIL) to evaluate the effectiveness of our proposed method. We begin by introducing the notations and formalism associated with CIL, followed by a definition of silent synapses and an exploration of their relevance to class-incremental learning.

### 2.1 CLASS-INCREMENTAL LEARNING

Class-Incremental Learning (CIL) refers to a scenario where a model retains previously acquired knowledge and continually learns to classify new classes, thereby constructing a unified classifier Rebuffi et al. (2017). Consider a sequence of classification tasks over $T$ datasets, denoted as $D^1, D^2, \ldots, D^T$, where $D^t = \{(x_i, y_i)\}_{i=1}^{n_t}$ is the $t$-th dataset $X_t$ containing $n_t$ instances. Each instance $x_i \in X_t$ is associated with a class label $y_i \in Y_t$, where $Y_t$ represents the label space of task $t$, and $Y_t \cap Y_{t'} = \varnothing$ for $t \neq t'$. In CIL, the objective is to build a unified classifier for all observed classes $\mathcal{Y}_t = Y_1 \cup \cdots \cup Y_t$ as the model is sequentially trained on datasets $\mathcal{X}_t = X_1 \cup \cdots \cup X_t$.

Our goal is to learn a model $M(x) : \mathcal{X}_t \to \mathcal{Y}_t$, which can be decomposed into two components: a feature embedding and a linear classifier. In the split learning setting, feature embedding is achieved using a pre-trained network $F(x) : \mathcal{X}_{t'} \to \mathcal{H}_{t'}$ and the $t$-th initialized sub-network $E_a^t(x) : X_t \to H_t$, which learns the feature embeddings for task $t$, where $t \neq t'$. In this study, we utilize a pre-trained Vision Transformer without its classifier, trained on a large dataset with strong zero-shot learning capabilities (Dosovitskiy, 2020; Smith et al., 2023; Zhou et al., 2024b). The $t$-th initialized sub-networks leverage Adapter modules to learn the features for different tasks, as proposed in (Chen et al., 2022). Following the settings in Zhou et al. (2024a;b), the linear classifier is defined by a function $l(\cdot) : \mathcal{H}_{t'} \cup H_1 \cup \cdots \cup H_t \to \mathcal{Y}_t$.

### 2.2 SILENT SYNAPSE IN BRAIN

Synapses are fundamental units of information storage in the brain, connecting pre-synaptic and post-synaptic neurons to facilitate learning plasticity and memory stability. During development, certain synapses are incapable of neurotransmission under basal conditions and remain in a reserve state until activated by an appropriate trigger Hanse et al. (2013). Silent synapses, which are prevalent in the hippocampus of young mammalian brains, exhibit an excitatory postsynaptic current that is absent at the resting membrane potential but becomes apparent upon depolarization. Functional synapses require both AMPA and NMDA receptors to activate connections, whereas silent synapses possess only NMDA receptors and lack AMPA receptors, rendering them inactive. Various studies have demonstrated that presynaptic activity, mimicked by glutamate uncaging, leads to the insertion of AMPA receptors in silent synapses. A current injection into the soma of the postsynaptic neuron produces a single action potential, thereby establishing new connections and converting silent synapses into functional synapses.

Silent synapses are abundant in the prepubescent brain, where they mediate circuit formation and refinement, but are thought to be rare in adulthood. Recently, Vardalaki et al. (2022) provided evidence that silent synapses on filopodia still occur in the adult brain. This discovery suggests a novel mechanism for the flexible control of synaptic wiring, potentially expanding the learning capabilities of the mature brain. Pre-trained networks in machine learning exhibit strong zero-shot learning capabilities, but their adaptation to new environments still requires continual learning. In this study, insights from silent synapses in the mature brain are employed to extend the class-incremental learning capabilities of pre-trained networks. We utilize artificial functional and silent synapses to create connections between the pre-trained network (memory stability in the mature brain) and the initialized sub-networks (learning plasticity in the mature brain).

## 3 PLASTICITY AND MEMORY STABILITY VIA SYNAPTIC CONSOLIDATION

In this section, we introduce the *Artsy framework*, wherein a pre-trained network connects to initialized sub-networks via silent and functional synapses for class-incremental learning tasks. We

first present the general network architecture of Artsy, followed by detailed training and inference algorithms leveraging synaptic consolidation.

## 3.1 MODEL ARCHITECTURE

The Artsy network architecture comprises four components: (1) a pre-trained network (analogous to the mature brain network, including the pre-synaptic neuron on the spine); (2) initialized sub-networks (analogous to the mature brain network, including the postsynaptic neuron on the filopodia); (3) artificial synapses (analogous to silent and functional synapses); and (4) a classifier.

**Pre-trained Network:** Models pre-trained on large datasets exhibit strong generalization capabilities. Similar to how young brains acquire extensive knowledge during development in dynamic environments, neurons with synapses in the mature brain mediate circuit formation and refinement to adapt to the real world. We posit that the pre-trained network is analogous to the adult brain. In machine learning, the pre-trained network encodes the features of new data $x$, expressed as:

$$h_0 = E_0(x), \tag{1}$$

where $h_0$ is the feature embedding of the new data produced by a pre-trained network $E_0(\cdot)$. The parameters of the pre-trained network remain fixed during the continual learning process, facilitating the learning of new tasks. Additionally, the mature functional synapses between neurons in the mature brain undergo minimal changes, thereby maintaining memory stability.

**Initialized Sub-networks:** Efficient fine-tuning is crucial for adapting pre-trained large models to downstream tasks. A popular approach involves using adapter methods, which initialize sub-networks with lightweight parameters. These methods achieve good performance on downstream tasks at a low cost. The adult brain also retains a capacity for neural plasticity and flexible, efficient learning, suggesting that the formation of new connections remains prevalent Vardalaki et al. (2022). This supports the idea that pre-trained networks can link to initialized sub-networks for efficient learning of new tasks. Furthermore, architecture-based continual learning methods often lack clear biological explanations Kudithipudi et al. (2022). In this study, biological evidence strongly supports the plausibility of architecture-based continual learning methods. The initialized sub-networks with trainable parameters learn the features of the new data $x_t$ in new $t$-th task, $t \in \{1, \cdots, T\}$, expressed as:

$$h_t = E_t(x_t), \tag{2}$$

where $h_t$ is the feature embedding of the new data by the sub-network $E_t(\cdot)$. This initialized sub-network can be seen as part of the new connections in the mature brain responsible for learning plasticity. However, establishing new connections between the initialized sub-network and the pre-trained network is crucial for balancing learning plasticity and memory stability. Incorrect connections between these networks can lead to erroneous predictions, akin to phenomena similar to illusions.

**Artificial Synapses:** Silent synapses are present not only in the developing brain Liao et al. (1995) but also in the mature brain Vardalaki et al. (2022), contributing to learning plasticity and memory stability. Inspired by the functionality and structure of silent and functional synapses, we propose artificial synapses to establish connections between presynaptic neurons in the pre-trained network and postsynaptic neurons in the initialized sub-networks. Artificial functional synapses directly connect presynaptic and postsynaptic neurons. Mimicking silent synapses, which possess NMDA receptors but lack AMPA receptors, a current injection into the soma of the postsynaptic neuron generates a single action potential, forming new connections. Artificial silent synapses require specific new stimuli to activate and convert into artificial functional synapses. We formally describe the artificial synapse as:

$$c_t = S_t \left( \sum_{i=0}^{t} h_i \right) = S_t \left( \sum_{i=0}^{t} E_i(x) \right), \tag{3}$$

$$m_t = \begin{cases} 0 & \text{if } c_t < \text{thr}_t, \\ 1 & \text{if } c_t \geq \text{thr}_t, \end{cases} \tag{4}$$

where different stimuli produce varying currents, leading to different silent synapses with distinct current thresholds to activate AMPA receptors. Here, $S_t(\cdot)$ is a function that calculates the related currents $c_t$ based on the new stimuli, ensuring $c_t$ is non-negative. An artificial synapse with a

---

**Algorithm 1 Artsy framework for Training CIL**

---

**Input:** Incremental traning datasets $D^T$, Incremental $T$ initialed sub-networks $E_t(x)$ and artificial synapses $S_t(x)$, $t \in \{1, 2, \cdots, T\}$; Pre-trained network: $F(x)$ via Eq 1.
**Output:** Incrementally trained model;
**for** $t = 1, 2, \cdots, T$ **do**
    Get a incremental training set $D^t$;
    Get a initialed new sub-network and artificial synapse: $E_t(x)$ via Eq 2 and $S_t(x)$ via Eq 3;
    Finetuning a new $E_t(x)$, Frozen old $\sum_{i=1}^{t-1} E_i(x)$, and new and old $\sum_{i=1}^{t} S_i(x)$;
    Optimize the networks: $\sum_{i=0}^{t} E_i(x)$;
    Complete the prototypes for former classes;
    Construct the prototypical classifier via Eq 5;
    Finetuning a new $S_t(x)$, Frozen old $\sum_{i=1}^{t-1} S_i(x)$, and new and old $\sum_{i=1}^{t} E_i(x)$;
    Optimize the networks: $S_t(\sum_{i=0}^{t} E_i(x))$;
    Test the model;
**end for**

---

threshold of $0$ is functional, while one with a threshold greater than $0$ is silent. By comparing $c_t$ with the threshold $\text{thr}_t$, we determine if the silent synapse activates ($m_t = 1$) or remains inactive ($m_t = 0$).

**Classifier:** We obtain the feature embedding of new data from the pre-trained network and the initialized sub-networks via artificial synapses. We then classify the new data using a non-parametric linear classifier, such as a prototype-based classifier Snell et al. (2017). Following the settings in Zhou et al. (2024a;b), the linear classifier is defined as:

$$y' = W^\top l \left( h + \sum_{i=1}^{t} m_i \cdot h_t \right), \tag{5}$$

where $y'$ represents the prediction, the classifier weights are denoted as $W = [\mathbf{w}_1, \mathbf{w}_2, \ldots, \mathbf{w}_{|\mathcal{Y}_b|}]$, and the weight for class $j$ is $\mathbf{w}_j$, and $l(\cdot)$ denotes the MLP-based layers used for extracting the final feature, identical to those in Zhou et al. (2024a).

### 3.2 SYNAPTIC CONSOLIDATION

Synaptic consolidation involves integrating new data while preserving existing memories during the training phase. Once synaptic consolidation is complete, the model is ready for inference during the testing phase.

**Training phase:** The training process for CIL follows several key steps. First, we establish $t$ incremental sub-networks and $t$ artificial synapses. For each incremental step $t$ from $1$ to $T$, we begin by acquiring the incremental training set. A new sub-network and artificial synapse are then initialized. The newly initialized sub-network is fine-tuned while keeping the previously trained sub-networks and all artificial synapses (both new and old) frozen.

Next, we optimize the combination of the pre-trained network $E_0(x)$ and all previously initialized and trained sub-networks, $\sum_{i=1}^{t} E_i(x)$, utilizing prior knowledge for learning new tasks. After completing the prototypes for the previous classes, we construct the prototypical classifier. The new artificial synapse is then fine-tuned, with the earlier artificial synapses and sub-networks remaining frozen. Finally, the networks are optimized, and the model is tested to evaluate its performance. The detailed procedure for the learning phase is outlined in Algorithm 1.

**Testing phase:** The testing process for CIL involves several key steps. Initially, we input the incremental testing datasets, $t$ trained sub-networks, $t$ trained artificial synapses, and the pre-trained network. We then acquire the incremental testing set. For each incremental step $t$ from $1$ to $T$, we retrieve and calculate the results $m_t$ using $S_t(x)$, where ($m_t \in 0, 1$). These results are stored in $R$. Next, we compile the results in $R = \{m_1, m_2, \cdots, m_T\}$. For each incremental step $t$, we obtain the feature embedding from $E_0(x) + \sum_{i=1}^{t} E^i(x) * m_i$ and input the feature embedding into the prototypical classifier. The detail of the test phase is shown in Algorithm 2.

---

**Algorithm 2 Artsy framework for Testing CIL**

---

**Input:** Incremental testing datasets: $D^t$; $t$ trained sub-networks and artificial synapses: $E_t(x)$ and $S_t(x)$, $t \in \{1, 2, \cdots, T\}$; Pre-trained network: $E_0(x)$;
**Output:** Incrementally tested model;
Get the incremental testing set $D$;
**for** $t = 1, 2, \cdots, T$ **do**
    Get a $S_t(x)$;
    Calculate the results $m_t$ via $S_t(x)$, where $m_t \in \{0, 1\}$;
    Store the results in $R$;
**end for**
Get the results in $R = \{m_1, m_2, \cdots, m_T, \}$;
**for** $t = 1, 2, \cdots, T$ **do**
    Get the feature embedding from $E_0(x) + \sum_{i=1}^t E_i(x) * m_i$ via Eq. 5;
    Input the feature embedding to the prototypical classifier;
**end for**

---

## 4 ARTSY AVOIDS CATASTROPHIC FORGETTING

In this section, we present the empirical evaluation of the Artsy framework on class-incremental learning tasks. First, we provide detailed descriptions of the experimental setup, including datasets, implementation details, and evaluation metrics. Next, we introduce the model parameters and the baseline methods used for comparison. Finally, we compare the performance of our method with state-of-the-art baselines on the CIFAR-100 and TinyImageNet datasets and conduct ablation studies to assess the impact of artificial synapses.

### 4.1 EXPERIMENTAL SETUP

We evaluate the performance of Artsy on class-incremental learning (CIL) tasks using the CIFAR-100 Krizhevsky et al. (2009) and TinyImageNet Yan et al. (2021) datasets. Furthermore, we conduct ablation studies to assess the contribution of artificial synapses to CIL tasks. The CIFAR-100 dataset, consisting of 100 classes, is divided into increments of 10 and 20 classes per step, with each class containing 500 training and 100 testing samples. TinyImageNet, a subset of ImageNet, comprises 200 classes, each with 500 training and 50 testing samples. In our experiments, we designate 100 classes of TinyImageNet as base classes for fine-tuning, while the remaining 100 classes are incrementally introduced in steps of 5, 10, and 20 classes per step. Following the methodology of Rebuffi et al. (2017), we evaluate Artsy and compare it with other baseline methods using two metrics: Average Accuracy ($\text{Avg}_t$) and Last Accuracy ($\text{Last}_t$). Here, $\text{Avg}_t$ denotes the mean Top-1 accuracy across all tasks up to task $t$, while $\text{Last}_t$ represents the Top-1 accuracy on the final task. For the $t$-th task, Average Accuracy is calculated as follows: $\text{Avg}_t = \frac{1}{T} \sum_{t=1}^T \text{Last}_t$.

### 4.2 ARTSY ARCHITECTURE AND BASELINES

We initialize the sub-network using AdaptFormer Chen et al. (2022) and construct the artificial synapse within Artsy using an MLP-based binary classifier. The network architecture and settings for AdaptFormer adhere to those described in Zhou et al. (2024b), with each AdaptFormer module dedicated to learning a new task. The MLP-based binary classifier, comprising two fully connected layers, is trained to classify data from previous tasks $\sum_{i=0}^{t-1} D^i$ as 0 and data from the current new task $D^t$ as 1. Consistent with Zhou et al. (2024b), all methods are implemented using the ViT-B/16 model as the pre-trained network. Experiments are conducted on NVIDIA 4090 and A100 GPUs, and all comparative methods are reimplemented in PyTorch. In Artsy, the sub-network is trained using the SGD optimizer with a batch size of 48 for 20 epochs. The artificial synapse is trained using the SGD optimizer with a batch size of 48 for 2 epochs, with the learning rate decayed from 0.01 using cosine annealing.

We compare our method with state-of-the-art CIL methods, including UCIR Hou et al. (2019), PASS Zhu et al. (2021), DER Yan et al. (2021), iCaRL Rebuffi et al. (2017), LwF Li & Hoiem (2017), and DyTox Douillard et al. (2022). Additionally, we compare our method with other CIL approaches that

Table 1: Comparison of Average and Last accuracies of Artsy and other Class-Incremental Learning methods on CIFAR-100 and TinyImageNet datasets. The best results are highlighted in grey.

| | | CIFAR-100 | | | | TinyImageNet | | | | | |
| | | 10 Steps | | 20 Steps | | 5 Steps | | 10 Steps | | 20 Steps | |
| Type | Name | Avg | Last | Avg | Last | Avg | Last | Avg | Last | Avg | Last |
|---|---|---|---|---|---|---|---|---|---|---|---|
| Regularization | iCaRL Rebuffi et al. (2017) | 79.35 | 70.97 | 73.32 | 64.55 | 77.02 | 70.39 | 73.48 | 65.97 | 69.65 | 64.68 |
| Regularization | LwF Li & Hoiem (2017) | 65.86 | 48.04 | 60.64 | 40.56 | 60.97 | 48.77 | 57.60 | 44.00 | 54.79 | 42.26 |
| Regularization | DER Yan et al. (2021) | 74.64 | 64.35 | 73.98 | 62.55 | - | - | - | - | - | - |
| Replay | UCIR Hou et al. (2019) | 58.66 | 43.39 | 58.17 | 40.63 | 50.30 | 39.42 | 48.58 | 37.29 | 42.84 | 30.85 |
| Replay | PASS Zhu et al. (2021) | - | - | - | - | 49.54 | 41.64 | 47.19 | 39.27 | 42.01 | 32.93 |
| Architecture | DyTox Douillard et al. (2022) | 67.33 | 51.68 | 67.30 | 48.45 | 55.58 | 47.23 | 52.26 | 42.79 | 46.18 | 36.21 |
| PTM | CLIP Thengane et al. (2022) | 75.17 | 66.72 | 75.95 | 66.72 | 70.49 | 66.43 | 70.55 | 66.43 | 70.51 | 66.43 |
| PTM | ZSCL Zheng et al. (2023) | 82.15 | 73.65 | 80.39 | 69.58 | 80.27 | 73.57 | 78.61 | 71.62 | 77.18 | 68.30 |
| PTM | MoE Yu et al. (2024) | 85.21 | 77.52 | 83.72 | 76.20 | 81.12 | 76.81 | 80.23 | 76.35 | 79.96 | 75.77 |
| PTM | EASE Zhou et al. (2024b) | 91.51 | 85.80 | 89.75 | 83.16 | 85.80 | 82.46 | 86.82 | 83.09 | 86.35 | 82.94 |
| | Artsy(Ours) | 92.44 | 87.94 | 90.54 | 85.09 | 87.30 | 84.59 | 87.79 | 84.19 | 87.86 | 84.21 |

utilize the same ViT-based pre-trained model, such as CLIP Thengane et al. (2022), ZSCL Zheng et al. (2023), MoE Yu et al. (2024), and EASE Zhou et al. (2024b).

## 4.3 RESULTS ON CIFAR-100 AND TINYIMAGENET

We report the Average Accuracy ($Avg_t$) and Last Accuracy ($Last_t$) results of our method, Artsy, and various class-incremental learning (CIL) baselines on the CIFAR-100 and TinyImageNet datasets, as presented in Table 1. Among the compared methods, UCIR Hou et al. (2019), PASS Zhu et al. (2021), DyTox Douillard et al. (2022), and DER Yan et al. (2021) are trained from scratch, while the remaining methods utilize the CLIP ViT-B/16 model as the backbone.

For CIFAR-100, Artsy achieves the highest Average and Last accuracy across 10 and 20 incremental steps, recording scores of 92.44% and 87.94% for 10 steps, and 90.54% and 85.09% for 20 steps, respectively. This performance surpasses all other methods, including those employing pre-trained models. On TinyImageNet, Artsy again demonstrates superior performance, achieving the highest scores across all evaluated steps. Specifically, for 5 steps, Artsy attains an Average accuracy of 87.30% and a Last accuracy of 84.59%; for 10 steps, the scores are 87.79% and 84.19%; and for 20 steps, they are 87.86% and 84.21%. These results indicate that Artsy consistently outperforms other traditional and pre-trained methods, such as CLIP Thengane et al. (2022) and ZSCL Zheng et al. (2023).

When compared with other pre-trained model (PTM)-based and architecture-based methods like EASE Zhou et al. (2024b) and MoE Yu et al. (2024), Artsy exhibits a clear advantage. For instance, on CIFAR-100, Artsy outperforms EASE by 0.91% in Average accuracy and 2.14% in Last accuracy for 10 steps, while surpassing MoE by 7.23% and 10.42%, respectively. On TinyImageNet, Artsy exceeds EASE by 1.5% in Average accuracy and 2.13% in Last accuracy for 5 steps, and outperforms MoE by 6.18% and 7.78%, respectively. These results demonstrate Artsy's robustness and effectiveness in handling class-incremental learning tasks. In conclusion, the results show that Artsy significantly enhances the performance of CIL tasks on both CIFAR-100 and TinyImageNet datasets. The integration of artificial synapses and the utilization of a pre-trained ViT-B/16 model contribute to its superior accuracy, making it a robust method for class-incremental learning.

## 4.4 LEARNING PLASTICITY AND MEMORY STABILITY IN ARTSY

We assess the learning plasticity and memory stability of Artsy and EASE on the CIFAR-100 dataset. Both methods are evaluated on a 10-step class-incremental learning task, assessing the accuracy on both newly introduced tasks and previously learned tasks at each incremental step. The accuracy on previously learned tasks quantifies the model's memory stability, whereas the accuracy on new tasks reflects its learning plasticity. Furthermore, we compute the Last Accuracy and Average Accuracy at each incremental step. The results, as depicted in Figure 3, demonstrate that Artsy surpasses EASE at most incremental steps, resulting in higher Average and Last Accuracy for Artsy at each stage. This indicates that Artsy effectively addresses the challenges inherent in continual learning by balancing learning plasticity and memory stability. The superior performance of Artsy underscores its

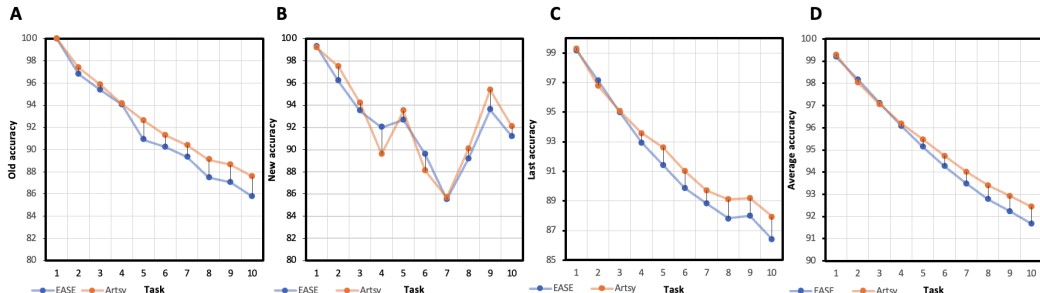

Figure 3: Comparison of learning plasticity and memory stability between Artsy and EASE. We evaluate the performance of Artsy and EASE on a 10-step class-incremental learning task using the CIFAR-100 dataset. The accuracy on both new and previously learned tasks is assessed at each step. Additionally, the final and average accuracies are computed at each step.

potential as a robust method for class-incremental learning, effectively retaining previously acquired knowledge while efficiently integrating new information.

### 4.5 ABLATIONS ON ARTSY

The Artsy framework comprises two primary components: (1) the network architecture, which includes a pre-trained network and multiple sub-networks, and (2) the artificial synapse, which connects the pre-trained network to the sub-networks. In our study, the sub-networks are implemented using adapter modules. These adapters exhibit strong capabilities for downstream tasks and have minimal impact on class-incremental learning (CIL) tasks Zhou et al. (2024a). Additionally, variants of pre-trained networks demonstrate strong zero-shot generalization capabilities for new tasks. Therefore, we focus our investigation on the artificial synapse rather than the network architecture in CIL tasks. The artificial synapse in Artsy is crucial for CIL tasks in this study. To effectively extract relevant information from test samples during inference, a well-designed artificial synapse is essential.

We focus on the ablation study of the artificial synapse, implemented as an MLP-based binary classifier. Different features extracted from new stimuli can influence the conversion of silent synapses into functional synapses. We compare the impact of different features (good feature and bad feature) from the pre-trained model on the activation of artificial silent synapses. As shown in Figure 4, using bad feature for activating the artificial synapse results in lower Last and Average accuracy compared to good feature. Bad feature cannot be effectively classified by the binary classifier, leading to incorrect connections between the networks and a failure to extract relevant information from the test samples. This experiment validates the importance of correct connections between the pre-trained network and sub-networks for CIL tasks. Furthermore, it highlights the necessity of selecting appropriate features for the artificial synapse and constructing an effective binary classifier to simulate artificial synapses. Furthermore, these findings support the hypothesis that brain lesions causing synaptic disconnections can lead to dementia by disrupting synaptic consolidation.

## 5 RELATED WORK

Continual learning methods are categorized into fixed-architecture and dynamic-architecture strategies. Fixed-architecture approaches, such as sequential learning, incrementally acquire new tasks while minimizing forgetting through regularization terms in the loss function Aljundi et al. (2018); Kirkpatrick et al. (2017); Li & Hoiem (2017); Ding et al. (2022), but may limit adaptability to novel tasks. Joint learning, another fixed-architecture method, employs replay of stored data or features from previous tasks Rebuffi et al. (2017); Hou et al. (2019); Buzzega et al. (2020); Zhu et al. (2021); Cha et al. (2021); Yan et al. (2021), balancing plasticity and stability but raising privacy concerns and storage issues. In contrast, dynamic-architecture (expansion-based) methods dynamically modify the network to incorporate new tasks while preserving prior knowledge Mallya & Lazebnik (2018); Serra et al. (2018); Wang et al. (2020); Douillard et al. (2022). While effective against catastrophic forgetting, they increase memory consumption and face challenges in task-agnostic settings.

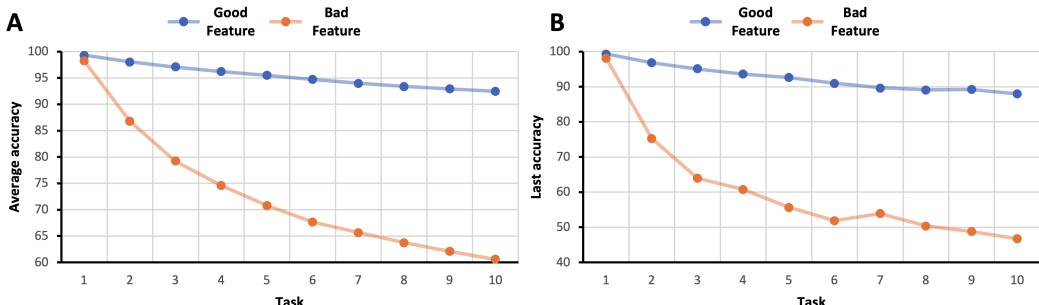

Figure 4: Ablation study results on the impact of good and bad features in activating the artificial silent synapse in Artsy for the 10-step task on CIFAR-100. Panels (A) and (B) depict the Average and Last accuracies at each step, respectively, utilizing the artificial synapse with distinct features.

Pre-trained model-based approaches leverage large-scale models' rich features for continual learning. Models like CLIP Radford et al. (2021); Thengane et al. (2022) and ZSCL Zheng et al. (2023) enhance performance across diverse tasks using pre-trained vision-language models. Adapters, lightweight modules enabling efficient adaptation of pre-trained models Houlsby et al. (2019), have gained traction in continual learning Ermis et al. (2022b;a); Liu et al. (2023); Yu et al. (2024) by adjusting limited parameters while retaining core feature extraction. An example is AdaptFormer Chen et al. (2022), offering efficient, lightweight parameterization to enhance model adaptability. This approach aligns with Vision Transformer-based continual learning frameworks like DyTox Douillard et al. (2022) and EASE Zhou et al. (2024b), facilitating new task learning while minimizing interference with prior knowledge. However, the limited capacity of adapter modules may restrict flexibility in handling diverse continual learning challenges.

## 6 CONCLUSION

Our methodology draws inspiration from the dynamic conversion of silent synapses into functional synapses, mirroring the adaptive learning mechanisms found in biological neural networks. This biologically inspired process enables the Artsy framework to strike an optimal balance between learning plasticity and memory stability, effectively addressing the challenge of catastrophic forgetting commonly observed in continual learning scenarios. Our findings suggest that integrating principles of synaptic plasticity into artificial neural networks can substantially improve their performance and resilience in lifelong learning tasks. Ablation studies further underscore the critical role of artificial synapses in maintaining both learning adaptability and memory stability. Specifically, constructing artificial networks that emulate the behavior of silent synapses is essential for distinguishing between previously learned data and novel data. Utilizing out-of-distribution detection methods Ran et al. (2022); Yang et al. (2024) can achieve this more effectively than employing an MLP-based binary classifier. Additionally, this approach offers potential avenues for simulating neurodegenerative conditions, particularly those marked by memory degradation due to synaptic dysfunction Forner et al. (2017). We also propose that Artsy can emulate synaptic consolidation, thereby opening new pathways for research into brain plasticity through the lens of artificial networks Albesa-González & Clopath (2024); Xu et al. (2024), with significant implications for both artificial intelligence and neuroscience.

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
