# OpenReview forum: "Brain-inspired continual pre-trained learner via silent synaptic consolidation"
_ICLR.cc/2025/Conference — ICLR 2025 Conference Withdrawn Submission_

### Official Review · Reviewer_dTcH · 2024-10-28

**Soundness:** 3
**Presentation:** 3
**Contribution:** 2
**Rating:** 5
**Confidence:** 4

**Summary:**

The authors introduced the Artsy framework, designed to enhance the continual learning capabilities of pre-trained models, addressing their vulnerability to catastrophic forgetting when incrementally trained on new tasks. Using their framework, the authors are able to achieve state-of-the-art performances on class-incremental learning tasks. Furthermore, this framework offered a promising avenue for simulating biological synaptic mechanisms.

**Strengths:**

1. The author's approach to constructing a pre-trained model inspired by the activation mechanisms of silent synapses is commendable.
2. The overall readability of the article is strong and easy to follow.
3. The use of artificial silent and functional synapses establishes precise connections between networks, enhancing the extraction of relevant information during inference.

**Weaknesses:**

1. The overall experiments have some shortcomings, as only two common datasets, CIFAR-100 and TinyImageNet, were used.
2. Although the authors emphasize biological synaptic mechanisms throughout the paper, corresponding results are not observed in the results section.
3. The authors mention that pre-trained artificial neural networks lack generalization capabilities, but they do not conduct corresponding experiments to address this issue.
4. We would have appreciated a more detailed exploration of how the authors intend to enhance the model based on synaptic mechanisms, accompanied by a mathematical description of these processes. Regrettably, the current explanation remains overly simplistic.

**Questions:**

1. Although the methods section of this article is described very clearly, many details are not introduced. For example, the pre-trained network E0(⋅).
2. The author needs to explain the mathematical mechanisms underlying artificial synapses.
3. The experiments are too weak, relying solely on two commonly used datasets (CIFAR-100 and TinyImageNet).
4. The author should include additional experiments that provide interpretability to highlight the advantages of the biological mechanisms.
5. The author should also provide some efficiency metrics to demonstrate the superiority of the model.

---

### Official Review · Reviewer_vQei · 2024-11-01

**Soundness:** 2
**Presentation:** 2
**Contribution:** 2
**Rating:** 3
**Confidence:** 4

**Summary:**

The authors proposed a continual learning system inspired by dynamical switching between silent and functional synapses in the brain. The actual mechanisms, however, has no real link to biological synapses that is extensively discussed in their study. Instead, their algorithm is a variation of the earlier algorithm referred to as ‘EASE’ in this study. EASE uses a pretrained encoder (visual transformer) as a backbone and trains adaptors to learn down-stream tasks. As each adaptor learns a new distinct task, catastrophic forgetting can be avoided.

**Strengths:**

The authors propose an automatic gating process that can turn on or off individual adaptors depending on present inputs. For each adaptor, a distinct MLP is trained to predict if a present input is an in-distribution example. During inference, adaptors are activated only when the corresponding MLPs predict “match”. Consequently, if all MLPs are perfectly trained and are 100% accurate, only a single adaptor trained for a present input will be activated, and other adapters will be shut down, which means that we can expect a highly accurate prediction.

**Weaknesses:**

This proposed gating process is interesting, but the authors’ own comparison to EASE show that its advantage is marginal. As they used two simple tasks (CIFAR100 and TinyImageNet) to evaluate the newly proposed algorithm, it remains unclear whether the proposed gating mechanism is beneficial for more complex tasks.

As MLPs need to be trained with old and new data, the algorithm proposed in this study requires a type of replay memory, which is not clarified in the paper.

The authors’ description of the model (e.g., encoder (E_t(x)), prototypes and MLP) also needs improvements. Their study is based on Zhou et al (2024) study that proposes EASE, so the details of their study may overlap with Zhou’s study, but this does not mean that they do not need to explain their algorithm. They should extend and improve the description of their proposed algorithm for better readability.

**Questions:**

Please see comments for "weaknesses".

---

### Official Review · Reviewer_k1G8 · 2024-11-01

**Soundness:** 2
**Presentation:** 2
**Contribution:** 3
**Rating:** 3
**Confidence:** 4

**Summary:**

This paper addresses the problem of catastrophic forgetting of pre-trained models by presenting an architecture for class-incremental learning.
The architecture, the Artsy framework, is inspired by the plasticity of neurons in the brain.
Artsy simulates silent and functional synapses. Specifically, (1) the fixed pre-trained network acts as a consolidated memory, (2) the sub-network learns the features of new incrementally available data, (3) artificial synapses interconnect the pre-trained network and sub-networks.
Here, artificial synapses represent the silent and functional synapses.
The experimental results show that Artsy achieves superior performance on incremental learning tasks.

**Strengths:**

**Originality:** An innovative biologically inspired approach to avoid catastrophic forgetting while using pre-trained models for incremental class learning is presented. The emulation of silent and functional synapses to artificial networks is appraising and novel in the context of pre-trained models.

**Quality:** A solid background on the biological foundations necessary to understand the composition of the Artsy framework is given.

**Clarity:** The flow of the text is easy to follow. The objective of the paper is clearly stated. All necessary background is given to understand the presented approach. The experimental setup is explained in detail.

**Significance:** The presented work is significant to advance the research in the domain of continuous/lifelong machine learning. The biological inspiration highlights this.

**Weaknesses:**

**Inconsistencies in formulas** There are some inconsistencies between equations and the presented algorithms. For example, LINE 230, Eq. (1) defines $h_0 = E_0(x)$, while LINE 272, Algorithm 1, uses $F(x)$ and LINE 326, Algorithm 2, uses $E_0(x)$.
LINE 323 states $E_0(x) + \sum_{i=1}^{t}E^i(x) * m_i$ which is different from the expression within the parentheses in Eq. (5). Equation (3) and (4) are not explained enough. For example, what is the purpose of $m_t$ in general (apart from determining whether a synapse is silent or functional) and how $c_t$ is learned?

**Weak ablation study**
The ablation study uses two different types of features as input to test the performance. For the ablation study per se, for example, it would be meaningful to see the separate contribution of $E_0(x)$ and $\sum_{i=1}^{t}E^i(x) * m_i$ to the performance on the class incremental learning task.

**Limited related work** While related work on silent and functional synapses and other approaches for class incremental learning is thoroughly presented, the related work on similar biologically inspired architectures is missing. Here, the comparison with other biologically inspired approaches would be beneficial. As an example, [1] can be considered.

[1] German I. Parisi, Ronald Kemker, Jose L. Part, Christopher Kanan, and Stefan Wermter. Continual lifelong learning with neural networks: A review. Neural Netw., 113(C):54–71, May 2019.

**Questions:**

1. How $thr_t$ is selected? Is it learnable?
2. What is $h$ in Eq. (5)? Is it $h_0$?
3. How is $S_t$ optimized (LINE 283)?
4. How often $m_t=0$ or $m_t=1$?
5.  What does `complete the prototypes for former classes' mean?
6. How was it determined that the sub-network is trained for 20 epochs and the artificial synapse is trained for 2 epochs?
7. What is a *good feature*? What is a *bad feature*?
8.  It would be interesting to see the performance on class incremental learning when only the pre-trained model is used. Are such experiments available?
9. Will the code be publicly available?

---

### Official Review · Reviewer_Xigk · 2024-11-03

**Soundness:** 2
**Presentation:** 2
**Contribution:** 2
**Rating:** 5
**Confidence:** 3

**Summary:**

The paper introduces the Artsy framework, which enhances continual learning in pre-trained models by mimicking the activation mechanisms of silent synapses via spike-timing-dependent plasticity observed in biological neural networks. The framework maintains memory stability for previously learned knowledge in the pre-trained network while promoting learning plasticity in task-specific sub-networks during training. During inference, it uses artificial silent and functional synapses to connect pre-synaptic neurons in the pre-trained network with post-synaptic neurons in the sub-networks, enabling effective information extraction. Experimental results show that Artsy outperforms conventional methods on class-incremental learning tasks and offers better biological interpretability than other solutions to mitigate catastrophic forgetting.

**Strengths:**

1. The paper presents an interesting and original idea by leveraging biological mechanisms of learning to enhance AI models, specifically focusing on the activation mechanisms of silent synapses through spike-timing-dependent plasticity. Given that the biological brain exhibits minimal effects of catastrophic forgetting compared to AI models, seeking inspiration from neurobiological learning mechanisms is a promising and innovative research direction. While initial explorations exist in the literature, there remains substantial room for further innovative research in this area.

2. Although some additional details are necessary for a complete explanation and reproducibility of the experiments, the authors have made a commendable effort in describing the framework by providing both training and inference algorithms and biological motivation.

3. The results, while needing a few more details for complete clarity, appear promising and suggest that the Artsy framework outperforms conventional methods on class-incremental learning tasks.

**Weaknesses:**

•	There are other literature works that propose biologically inspired solutions to mitigate catastrophic forgetting (see below), which are not covered in the background nor related work sections. I suggest adding these references to provide a more comprehensive context for the proposed framework in the neuroscientific context.

•	Some parameters needed for reproducibility of the results (incl. number of parameters, type of connectivity e.g. for silent and functional synapses) are not reported. For instance, the paper does not mention how many artificial synapses are used for each subnetwork or whether there is e.g. all-to-all vs sparse connectivity.

•	The study mentions limitations of other algorithms in the background section regarding efficiency and computational time. However, the authors do not discuss these features of the Artsy framework compared to other algorithms. Efficiency, model complexity and computational time are important aspects that the authors should quantitatively analyze (or at least provide estimates for) to explain the performance vs efficiency tradeoff.

•	A potential limitation of the framework is the potential increase in the number of subnetworks and artificial synapses with the addition of more classes. This could pose scalability issues and raise questions about the biological plausibility of the framework. I suggest that the authors provide more information and comments on this aspect.

•	The paper does not provide a link to the code, which is essential for reproducibility and further validation of the results.

•	Standard deviations for the results in Table 1 and Figure 3 and 4 are not shown. These are important for comparing the variability of the frameworks. In addition, the experimental setup lacks clarity regarding the number of runs averaged (for instance, for Figure 4B).

•	“Good” and “bad features” (sec 4.5) are not clearly defined

•	The paper makes a claim that brain lesions causing synaptic disconnections can lead to dementia by disrupting synaptic consolidation but lacks references to support this. Moreover, the connection between artificial synapses of the Artsy framework and brain lesions needs to be made clearer.

•	More targeted explanations of AMPA and NMDA receptors are needed, for example if the relevance of these receptors to short- vs long-term plasticity is related to the dynamics of functional and silent synapses

•	The diagram for Figure 2C could show more than one subnetwork to accurately represent the architecture.

If the above points are clarified, I am happy to revise my score.

Suggested references (non exhaustive):
https://arxiv.org/pdf/2405.09637 , https://www.nature.com/articles/s42256-023-00747-w , https://www.nature.com/articles/s41467-022-29491-2 , https://arxiv.org/pdf/2403.13249 , https://proceedings.mlr.press/v232/madireddy23a.html , https://pubmed.ncbi.nlm.nih.gov/37589728/ , https://proceedings.mlr.press/v162/gurbuz22a/gurbuz22a.pdf

**Questions:**

Minor:
1.	The type function used for S_t is not specified. I suppose it is a step function, if so could you please confirm and specify?

2.	Where does the name "Artsy" originate from? Is it an abbreviation for "ARTificial Synapse"? If so, could you specify this in the paper?

3.	Can the Artsy framework work in other continual learning settings beyond class-incremental learning (CIL)?

4.	How are both the pre-trained network and the initialized subnetworks analogous to the mature brain network? Could you provide examples and references of networks and subnetworks coexisting and connected in the mature brain, but with different dynamics?

5.	What is the rationale behind naming the connections "artificial synapses"?

---

### Note · Authors · 2024-11-24

I have read and agree with the venue's withdrawal policy on behalf of myself and my co-authors.